# De Novo Transcriptome Profiling of *Naegleria fowleri* Trophozoites and Cysts via RNA Sequencing

**DOI:** 10.3390/pathogens12020174

**Published:** 2023-01-22

**Authors:** Hae-Jin Sohn, Jong-Hyun Kim, Kyongmin Kim, Sun Park, Ho-Joon Shin

**Affiliations:** 1Department of Microbiology, Ajou University School of Medicine, Suwon 16499, Republic of Korea; 2Department of Biomedical Science, Graduate School of Ajou University, Suwon 16499, Republic of Korea; 3Institute of Animal Medicine, College of Veterinary Medicine, Gyeongsang National University, Jinju 52828, Republic of Korea

**Keywords:** *Naegleria fowleri*, cysts, trophozoites, RNA sequencing, transcriptome

## Abstract

*Naegleria fowleri* is a pathogenic free-living amoeba, commonly found around the world in warm, fresh water and soil. *N. fowleri* trophozoites can infect humans by entering the brain through the nose and causing usually fatal primary amebic meningoencephalitis (PAM). Trophozoites can encyst to survive under unfavorable conditions such as cold temperature, starvation, and desiccation. Recent technological advances in genomics and bioinformatics have provided unique opportunities for the identification and pre-validation of pathogen-related and environmental resistance through improved understanding of the biology of pathogenic *N. fowleri* trophozoites and cysts at a molecular level. However, genomic and transcriptomic data on differential expression genes (DEGs) between trophozoites and cysts of *N. fowleri* are very limited. Here, we report transcriptome Illumina RNA sequencing (RNA-seq) for *N. fowleri* trophozoites and cysts and de novo transcriptome assembly. RNA-seq libraries were generated from RNA extracted from *N. fowleri* sampled from cysts, and a reference transcriptome was generated through the assembly of trophozoite data. In the database, the assembly procedure resulted in 42,220 contigs with a mean length of 11,254 nucleotides and a C+G content of 37.21%. RNA sequencing showed that 146 genes in cysts of *N. fowleri* indicated 2-fold upregulation in comparison with trophozoites of *N. fowleri*, and 163 genes were downregulated; these genes were found to participate in the Kyoto Encyclopedia of Genes and Genomes (KEGG) pathway. The KEGG pathway included metabolic (131 sequences) and genetic information processing (66 sequences), cellular processing (43 sequences), environmental information processing (22 sequences), and organismal system (20 sequences) pathways. On the other hand, an analysis of 11,254 sequences via the Gene Ontology database showed that their annotations contained 1069 biological processes including the cellular process (228 sequences) and metabolic process (214 sequences); 923 cellular components including cells (240 sequences) and cell parts (225 sequences); and 415 molecular functions including catalytic activities (195 sequences) and binding processes (186 sequences). Differential expression levels increased in cysts of *N. fowleri* compared to trophozoites of *N. fowleri*, which were mainly categorized as serine/threonine protease, kinase, and lipid metabolism-related proteins. These results may provide new insights into pathogen-related genes or environment-resistant genes in the pathogenesis of *N. fowleri*.

## 1. Introduction

*Naegleria fowleri* is a ubiquitous pathogenic free-living amoeba, colloquially known as the ‘brain-eating amoeba’, and causes primary amoebic meningoencephalitis (PAM) [1,2,3,4]. *N. fowleri* has three types of life stage: cyst, trophozoite, and flagellate. The amoebae are found in rivers, ponds, lakes, and swimming pools. When contaminated water enters through the nose and passes through the cribriform plate in the brain, *N. fowleri* infection is the most frequent result. *N. fowleri* infection is an extremely rare and severe fatal brain infection, which usually occurs in children and young adults who engage in water activities. Infections with *N. fowleri* are rare, but they occur mainly during the summer and are reported each year [5,6,7].

The clinical signs of PAM are similar to those of bacterial and viral meningitis, including fever and severe headaches. Because of the difficulty in initial detection, about 75% of diagnoses are made after the death of the patient [7,8]. *N. fowleri* infection is a rare and serious infection of the brain. There is no clear treatment or diagnosis using specific laboratory test in laboratories. In spite of its medical, public health, and environmental importance, the genetics and biology of this amoeba remain poorly understood.

Recent development in next-generation sequencing (NGS) technology, such as Illumina platforms, has dramatically improved the efficiency of gene discovery. RNA sequencing (RNA-seq) is an NGS method for profiling transcripts using deep sequencing technology [9,10,11]. In particular, the de novo assembly does not require a reference genome and is used when the sequence is unknown or incomplete. It goes through the process of analyzing and mapping the base sequence to form a genome and transcriptome [12,13]. 

To date, many parasites transcriptomes have been sequenced using NGS technology, such as *Acanthamoeba* spp. [14,15]; *Entamoeba histolytica* [16,17,18,19]; *Plasmodium* spp. [20]; *Clonorchis sinensis* [21,22,23]; *Schistosoma* spp. [24,25,26]; and *Fasciola hepatica* [27,28,29]. To obtain transcriptomic insights into the differential expression of the stage-specific gene in *N. fowleri*, this study used RNA-Seq to analyze the transcriptome of *N. fowleri* cysts transferred to encystment buffer for 3 days compared with an *N. fowleri* trophozoite control group. In addition, this study was designed to produce transcriptomic data to aid in better understanding the biology of *N. fowleri*, which would facilitate the identification of pathogenic-related genes or environment-resistant genes in *N. fowleri* infection.

## 2. Materials and Methods

### 2.1. Cultivation of N. fowleri and RNA Isolation 

*N. fowleri* trophozoites (Carter NF69; ATCC No. 30215) were axenically cultured at 37 °C in Nelson’s medium containing 10% fetal bovine serum [30]. To induced encystation, *N. fowleri* trophozoites were transferred into an encystation buffer (120 mM NaCl, 0.03 mM MgCl_2_, 1 mM NaHPO_4_, 1 mM KH_2_PO4, 0.03 mM CaCl_2_, 0.02 mM FeCl_2_, pH 6.8) [31]. After 72 h of incubation in encystation medium, the amoebae were harvested via centrifugation at 1500 rpm for 3 min at room temperature. The total RNA of trophozoites and cysts was isolated using RNeasy kits (QIAGEN, Hilden, Germany) according to the manufacturer’s protocol. The integrity and purity of the extracted total RNA were assessed using the NanoDrop 1000 (Thermo Fisher Scientific, Waltham, MA, USA) and Bioanalyzer 2100 systems (Agilent Technologies, Santa Clara, CA, USA).

### 2.2. Sequencing and De Novo Transcriptome Assembly 

A total amount of 1 µg RNA (A260/A280 ratio ≈ 2.0–2.18, A260/A230 ≈ 2.27–2.37 and RIN > 7) was used for the construction of the cDNA library. cDNA was synthesized using mRNA fragments as templates. mRNA was selected using oligo dT probes, and then, fragmented using divalent cations. The short fragments were purified and resolved using an EB buffer for end reparation and single-nucleotide adenine addition. Afterwards, the short fragments were connected with adapters. The library was loaded using an Illumia HiSeq 2500 instrument for 10 gigabase in-depth sequencing. Raw sequence reads were quality assessed using FastQC (http://www.bioinformatics.babraham.ac.uk/projects/fastqc/; accessed on 19 January 2023) followed by quality and adapter trimming using the Trinity assembler with default parameters. All sequence data approaches and expression profiles of each gene at stage-specific genes were determined.

### 2.3. Functional Annotation 

Differentially expressed genes between *N. fowleri* trophozoites and cysts samples were determined using DEseq with the cutoff for fold change set at >2 and for false discovery rate (FDR) at <0.05. For the second approach, a transcript dataset of *N. fowleri* was constructed. Briefly, all the clean reads were assembled using Trinity (version 2.0.2, http://trinityrnaseq.github.io/; accessed on 19 January 2023 [32]), and transcript datasets of *N. fowleri* were constructed using the above approaches. Functional annotation was performed using BLAST+/SwissProt, Gene Ontology (GO), and Kyoto Encyclopedia of Genes and Genomes (KEGG) gene sequences, and related parameters were analyzed using the appropriate software.

### 2.4. Identification of Stage-Specific Genes 

The expression levels of transcripts from the *N. fowleri* trophozoite and cyst libraries were calculated by mapping them to the reference transcriptome that was created via the de novo assembly mentioned above. We followed RSEM (v1.2.29) to assign reads to expression levels via abundance estimation using FPKM (fragments per kilobase of transcript per million fragments mapped). We performed the differential gene expression analysis using a matrix file containing the mapped read counts for *N. fowleri* trophozoites and cysts. We searched for stage-specific related genes in the finalized transcriptome of *N. fowleri* using the following parameters: an e-value < 1e − 10, an FPKM > 1, and subject coverage > 60%. 

### 2.5. Reverse-Transcription PCR

To assess the expression levels of the profilin gene by *N. fowleri* trophozoites and cyts, the total RNA of *N. fowleri* was isolated using an RNeasy^®^Mini kit (QIAGEN, Valencia, CA, USA). The cDNA was prepared from 5 μg of total RNA in a reaction mixture containing oligo (dT) primers. The mixtures were reacted at 42 °C for 1 h, then, for 5 min at 94 °C and at 4 °C. RT-PCR was performed using the profilin gene-specific primers. The PCR conditions were as follows: 95 °C for 5 min, 30 cycles at 95 °C for 30 s, 55 °C for 30 s, 72 °C for 1 min, and then, a final extension for 10 min at 72 °C. On 1% agarose gel, the PCR products were separated and stained with ethidium bromide.

## 3. Results

### 3.1. Sequencing and De Novo Assembly

To investigate stage-specific gene expression levels, we produced Illumia RNA-seq libraries from trophozoites for *N. fowleri*. Deep sequencing of the libraries yielded about 140 million paired-end reads that were combined, quality filtered, and de novo assembled using the Trinity software. Trinity assembler was then used to pool and assemble the clean reads into 13,400 contigs with a total length of 33,118,105 bp, a mean length of 1180.5, a weighted median length (N50) of 5360 bp, and GC content of 37.21%. The transcriptome assembly from the cDNA library of the *N. fowleri* trophozoites and cysts achieved a quality of ORF/coding sequence for analysis (Table 1, Figure 1)

### 3.2. Transcriptome Annotation

The sequences were mapped to Gene Ontology (GO) terms and subsequently annotated. A total of 11,254 (total trinity genes) sequences were annotated with GO terms in the *N. fowleri* trophozoite and cyst transcriptome. To select the significant GO terms, a *p*-value cutoff of 0.005 was used. The GO terms were grouped according to biological processes, molecular functions, and cellular components. The 2407 sequences were analyzed using the GO database (Figure 2). In the biological process, 1069 sequences were expressed including cellular process (228 sequences) and metabolic process (214 sequences). In the cellular component, 923 sequences were expressed including cells (240 sequences) and cell parts (225 sequences). In molecular function, 415 sequences were expressed, and the most frequently expressed sequences included catalytic activity (195 sequences) and binding (186 sequences). 

A total of 11,254 sequences were mapped, with 282 mapped to KEGG pathways (Table 2). A significant proportion of amino acid sequences were associated with (1) metabolic pathways (131 sequences) including lipid metabolism, energy metabolism, the metabolism of other amino acids, nucleotide metabolism, carbohydrate metabolism, amino acid metabolism, the metabolism of terpenoids and polyketides, xenobiotic biodegradation and metabolism, the metabolism of cofactors and vitamins, glycan biosynthesis and metabolism, and the biosynthesis of other secondary metabolites; (2) genetic information processing (66 sequences) including translation, folding, sorting and degradation, replication, and repair; (3) cellular processes (43 sequences) including cell motility, cell growth and death, cellular community, transport, and catabolism; (4) environmental information processing (22 sequences) including signal transduction, membrane transport, signaling molecules, and interaction; and (5) organismal systems (20 sequences) including the immune system, the digestive system, the nervous system, development, and the endocrine system (Table 2).

### 3.3. Differential Expression Genes (DEGs)

To identify the DEGs of *N. fowleri* trophozoites and cysts for each gene, log Fc was calculated, and DEGs including an adjusted *p* < 0.05 and log-2-fold change ratios > 1 were selected (Figure 1). We analyzed in the transcriptome database the expression of 38 genes related to cytoskeleton protein, 24 genes encoded for cell growth and death, and 49 genes potentially implicated in signal transductions. A total of 10 and 14 upregulated DEGs in *N. fowleri* trophozoites and cysts, respectively, are shown in Table 3. In this study, we focused on the expression of the profilin gene among various genes expressed in *N. fowleri* trophozoites and cysts. Because the profilin gene regulates the nucleation rate of actin polymerization in other organisms, including *Acanthamoeba* [33], and the rate of filament elongation, and reduces the concentration of F-actin, profilin is also assumed to play an important role in the pathogenesis of *N. fowleri*. Results from the gene expression analysis showed that profilin (log Fc = 6.814, *p* < 0.001) was differentially expressed in the cyst stage of *N. fowleri* (Table 3). The profilin gene, in particular, exhibited the fifth highest expression level during the cyst stage, while there was no expression level in trophozoite stage of *N. fowleri*. 

### 3.4. Gene Expression of Profilin in N. fowleri Trohozoites and Cysts

To confirm the expression of the profilin gene in *N. fowleri* trophozoites and cysts, RT-PCR analysis was carried out (Figure 3). We selected a primer to clone the profilin gene found by the Blast search (Appendix A). To perform RT-PCR analysis, we synthesized cDNA from total RNA extracted from *N. fowleri* trophozoites and cysts. The expression of profilin was observed at 450 bp in *N. fowleri* cysts but not in *N. fowleri* trophozoites (Figure 3).

## 4. Discussion

Pathogenic *N. fowleri* resides in various environments and causes a brain infection called PAM. It is unclear which genes are expressed in each life stage that cause brain infection. A detailed understanding of the information on the differential expression genes between the trophozoites and cysts of *N. fowleri* may provide novel pathogenic factors in amoeba infection. Transcription profiling provides information on and fundamental insights into biological processes. In this study, the transcript analysis of *N. fowleri* trophozoites and cysts via RNA-seq was performed. We used a high-throughput sequencing system to profile the *N. fowleri* trophozoite and cyst transcriptome using the Illumina HiSeq 2500 system. Through DEGs analysis, we obtained various genes involved in *N. fowleri* during the cyst stage.

We assembled transcriptomes from trophozoites and cysts and identified a total of 11,254 genes with an average length of 2471.50 bp. A total of 42,220 genes were annotated using four major databases. We found 146 DEGs in cysts and 163 DEGs in trophozoites (2-fold expression, FDR < 0.5). The transcriptomic genes related to cellular motility, growth and death, signal transduction, translation, carbohydrate metabolism, lipid metabolism, and nucleotide metabolism showed various expression levels during cyst and trophozoite formation. These results related to amoebae proliferation, differentiation, and growth/death. GO and pathway analysis of the DEGs showed that ‘cells’, ‘cellular processes’, and ‘catalytic activity’ were the most increased categories. These findings will be useful in further studies of the other pathogenesis-related gene in *N. fowleri*.

Our study also showed increased expression of many genes encoding cytoskeleton-related genes such as actin, tubulin, myosin, and actin-binding proteins in *N. fowleri* trophozoites. A total of 38 genes were associated with cytoskeleton-related protein. These genes could be classified into six GO categories and three pathways. In the ‘cellular processes’ pathway, genes were mainly associated with cell motility, growth/death, and cellular community. In the ‘environmental information processing’ pathway, genes were related to signal transduction such as serine threonine-protein kinase, Rho GTPase, and tyrosine kinase. Moreover, these pathways provide a starting point for exploring genes related to amoeba development and pathogenesis and understanding their molecular functions.

There are many reports of the function of the cytoskeleton involved in pathogenicity in protozoa such as *E. histolytica* [34,35,36], *T. gondii* [37,38], and *Acathamoeba* spp. [39,40]. Recently, studies of the transcriptome of *Balamuthia mandrillaris* trophozoites for structure-guided drug design and treatment with Hesperidin conjugated with silver nanoparticles using RNA-Seq in *N. fowleri* infection were reported [41,42]. In a previous study, Nf-actin was located in the cytoplasm, pseudopodia, and amoebastome in *N. fowleri* trophozoites [43]. *N. fowleri* that was overexpressed Nf-actin showed significantly increased adhesion, phagocytic activity, and cytotoxicity [44].

In particular, levels of the profilin gene in cysts were significantly increased, whereas those in trophozoites were not changed. It is reported that the profilin gene regulates the nucleation rate of actin polymerization and the rate of filament elongation and reduces the concentration of F-actin [45,46]. Moreover, some genes such as cytoskeletal genes and pathogenesis-related genes, and the heat shock protein, showed increased expression levels in the cyst stage compared with the *N. fowleri* trophozoites stage. These results indicate that cytoskeleton-related genes play important roles in the pathogenesis of *N. fowleri* infection.

Although the pathways used by *N. fowleri* cysts to access the host are poorly understood, our data suggest that this step could be facilitated by transporters that are upregulated in our transcriptome. Even though the serine/threonine protease, kinase, and lipid metabolism-related proteins are still unknown, they have been suggested to act as regulators of cyst formation in *N. fowleri*. Thus, these results could indicate that stage-specific genes may be important regulators in the pathogenesis of *N. fowleri* infection.

In conclusion, we used RNA-seq to identify the various DEGs in *N. fowleri* trophozoites and cysts. Additionally, we obtained stage-specific genes that were significantly upregulated for each stage. This information can be utilized to assess developmental competence concerning cellular processes, metabolism, and the immune system in amoeba–host interaction, which could be associated with brain infection. Additionally, these results could be valuable transcriptomic/genomic resources for further analysis of genes involved in cellular mechanisms and the immune system in other protozoan infections.

## Figures and Tables

**Figure 1 pathogens-12-00174-f001:**
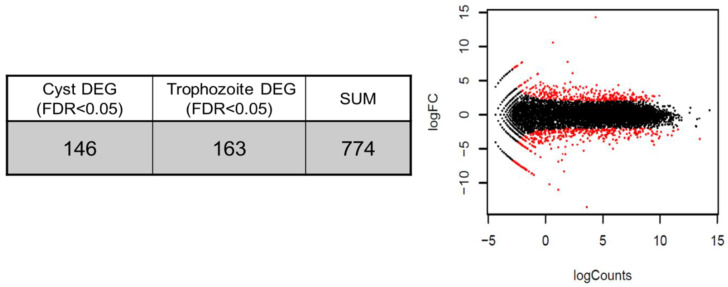
Differential expression analysis of the N. fowleri trophozoites and cysts. The assembly procedure results in 42,220 contigs with a mean length of 11,254 nucleotides and a C+G content of 37.21%. The cysts’ (146) and trophozoites’ (163) differential expression genes (DEGs) were analyzed using the Gene Ontology (GO) data base. SUM: total of cyst and trophozoite DEGs, black dot: total read count, red dot: cyst and trophozoite DEGs.

**Figure 2 pathogens-12-00174-f002:**
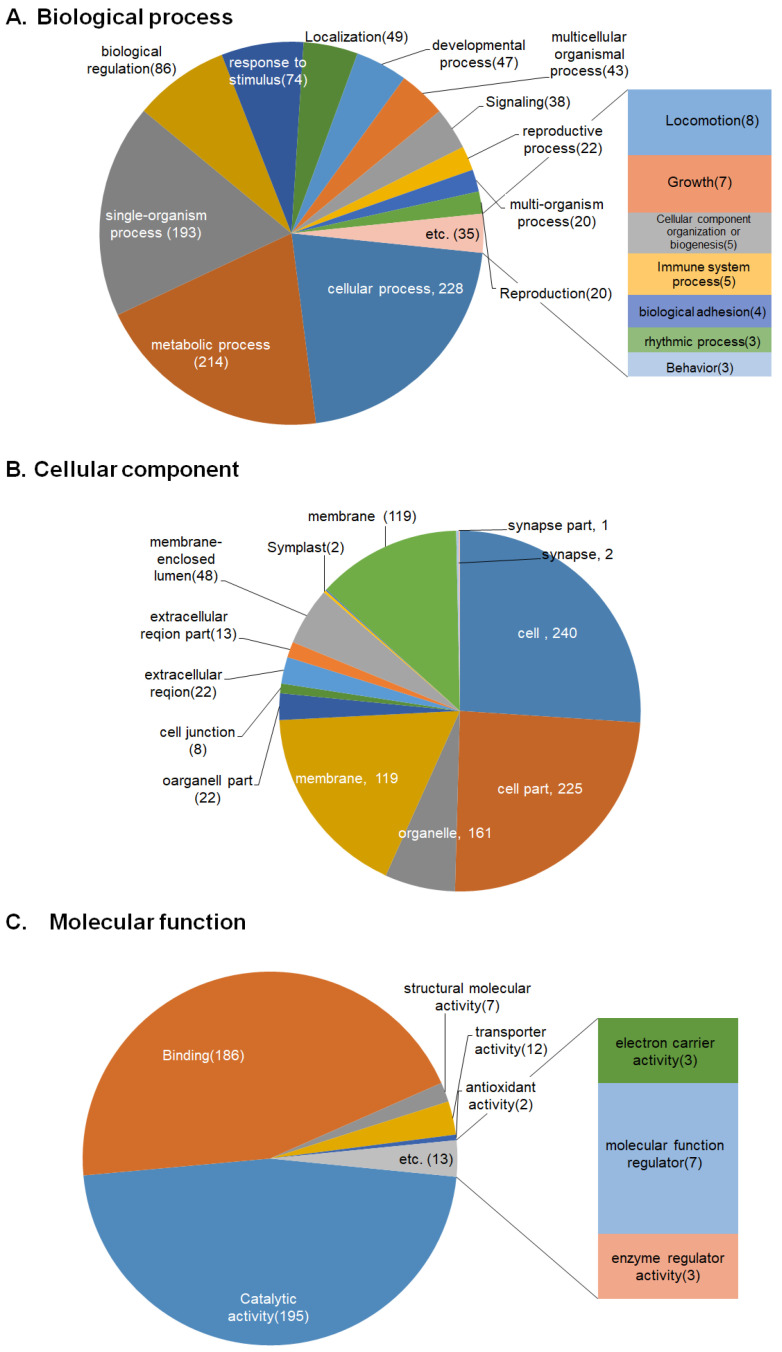
Functional annotations of the *N. fowleri* trophozoites and cysts based on Gene Ontology (GO) categories. The pie charts show the general categories of biological process (**A**), cellular component (**B**), and molecular function (**C**).

**Figure 3 pathogens-12-00174-f003:**
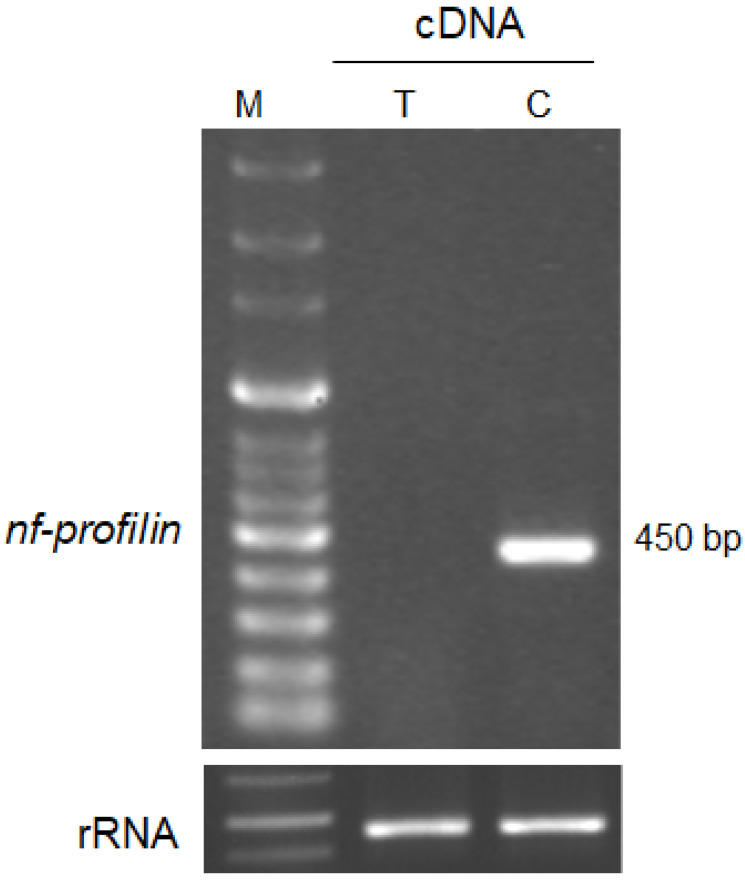
Expression levels of *nf-profilin* and control *(N*. *fowleri* ribosomal RNA sequence) in *N. fowleri* trophozoites and cysts (M: marker, T: *N. fowleri* trophozoites, C: *N. fowleri* cysts).

**Table 1 pathogens-12-00174-t001:** Characteristics of the transcriptome of the *N. fowleri* trophozoites and cysts.

Contig Count	42,220
**Type**	De novo assembly
**Total read count**	135,733,193
**Mean read length (bp)**	1180.5
**Total read length (bp)**	135,733,193
**Mean contig length (bp)**	2471.5
**Total contig length (** **bp)**	33,118,105
**Contig N50 value (bp)**	5360
**GC content (%)**	37.21

**Table 2 pathogens-12-00174-t002:** Pathway in *N. fowleri* trophozoites and cysts mapped by the Kyoto Encyclopedia of Genes and Genomes (KEGG).

Pathway	
**Cellular Processes**	**43**
Cell motility	22
Cell growth and death	10
Cellular community	4
Transport and catabolism	7
**Environmental Information Processing**	**22**
Signal transduction	12
Membrane transport	7
Signaling molecules and interaction	3
**Genetic Information Processing**	**66**
Translation	31
Folding, sorting, and degradation	23
Replication and repair	12
**Metabolism**	**131**
Lipid metabolism	22
Energy metabolism	11
Metabolism of other amino acids	9
Nucleotide metabolism	19
Carbohydrate metabolism	37
Amino acid metabolism	8
Metabolism of terpenoids and polyketides	3
Xenobiotic biodegradation and metabolism	6
Metabolism of cofactors and vitamins	8
Glycan biosynthesis and metabolism	2
Biosynthesis of other secondary metabolites	6
**Organismal Systems**	**20**
Immune system	5
Digestive system	5
Nervous system	4
Development	1
Endocrine system	5

**Table 3 pathogens-12-00174-t003:** The list of genes that showed upregulated proteins in *N. fowleri* trophozites and cysts.

Upregulated Proteins in *N. fowleri* Trophozoites	log FC
Luminal-binding protein	14.309
Lysosomal Pro-X carboxypeptidase	7.017
Chaperone protein	4.349
12-oxophytodienoate reductase 1	4.344
Probable alpha-L-glutamate ligase	4.181
fatty acid desaturase	4.120
Cytoskeleton-associated protein	3.280
Tubulin alpha-6 chain	2.659
Actin	2.115
Microtubule-associated protein	1.996
**Upregulated Proteins in *N. fowleri* Cysts**	**log FC**
Uridine kinase	13.526
Calpain-5	8.641
Translin-associated factor X-interacting protein	7.394
Gag-Pol polyprotein	6.912
Profilin	6.813
Probable E3 ubiquitin-protein ligase	6.707
Serine/threonine-protein kinase	6.683
LisH domain-containing protein	5.032
Phospholipid-transporting ATPase	4.970
EF-hand domain-containing family member C2	4.579
Kinesin-like calmodulin-binding protein	4.429
Probable glycerol-3-phosphate dehydrogenase(mt)	4.327
Sphingosine-1-phosphate lyase	4.285
Nitrile-specifier protein	4.174

## Data Availability

The data supporting the conclusions of this article are provided within the article. The original data in the present study are available from the corresponding authors.

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
