# Peer review of "De Novo Transcriptome Profiling of Naegleria fowleri Trophozoites and Cysts via RNA Sequencing"

_pathogens, 2023, doi:10.3390/pathogens12020174_

Round 1

Reviewer 1 Report

 I think that this study "De novo transcriptome profiling of Naegleria fowleri trophozoites and cysts by RNA-sequencing" has a very potential interest regarding the description of N. fowleri (trophozoite or cyst) transcriptomic. Moreover, the authors use methods with deep sequencing technology which would help to understand the pathogenesis of the amoeba.  Some questions, observations and recommendations, are found below. 

MATERIAL AND METHODS

LINE 86: 1microg, looks "u" instead of "m"

RESULTS

Table No 1 add bp or genes as appropriate after the numbers

Figure 1. Trophozoites and cysts have the same DEG, I don't understand the distribution of the values in terms of cysts and trophozoites from the small table. Which value corresponds to trophozoites and which to cysts? 

Furthermore, these values do not match with the values of line 212 of the discussion, which I think must correspond to the same values.

Line 146. I think the total is 2407 instead 2408 (415+1069

923)

Lines from 143 to 152. Could you describe the results as you are representing in fig. 2. Same order. First Biological process, cellular component and finally molecular function.  To follow better the description..

Table 2. Could you explain better how this section is built, I am supposed to observe trophozoites and cysts with different expressed sequences

Line 174 is p<0.005 or p<0.05? as you are representing in figure 1

Line 178. After cysts add the word "respectively". Same line, All these sequences  (10 of trophozoites and 14 of cysts) were considered as responsive to locomotion? Could you explain in the discussion how some of them are participating in cysts?

Line 184 a reference is required. Moreover, if profilin is important in the pathogenesis, why trophozoites did not upregulate it (table 3)

The cyst can infect? Is there any report about cyst invasion and consequently infection?  

In line 185 authors mention that "Results from the gene expression analysis showed that profilin (log Fc=6.814, p < 0.001) 185 was differentially expressed in the cyst stage of N. fowleri (Table 3). Especially the profilin  gene exhibited the highest expression level during the cyst stage, while the expression 187 level in trophozoites stage of N. fowleri was lowest"  

The table shows log FC of 6.813, you have to correct it and where we can find or observe that profilin had the highest expression value as the authors describe. Profilin had the highest expression level in cysts?

DISCUSSION

Line 211, 11,254 genes do not match with 11,354 that you describe in the results  (line 143)

Line 212. The values also do not match with fig 1 of the results 

From lines, 213 to 216  authors describe that "The transcriptomic genes related to cellular motility, growth and death, signal transduction, translation, carbohydrate metabolism, lipid metabolism and  nucleotide metabolism showed various expression levels during cyst formation

I think that they refer to those found in table 2; however, where it is observed.  I can not observe a cyst section for expression levels 

Many lines of discussion consider that would be better in the results sections (lines 210 to 227)

Line 218, explain better this idea. How?

Line 227. Specifically, why this pathway provides a starting for studying pathogenesis. I think you can develop better the idea. 

Reviewer 2 Report

Overall, the manuscript is well written. M&M could be improved. In addition, the results show the importance of differences between trophozoites and cysts in the expression of N. fowleri-related genes, but I think the data provided could be discussed a bit more.

Minor changes:

-Line 14: change "fetal" for "fatal" (PAM)

-Line 64: change "Acanthameba" for "Acanthamoeba" 

-Line 192: Which primers were used exactly? Refers them properly. 

-Recomendation: Change references to more current and updated studies in general.

Reviewer 3 Report

Sohn et al 2023 De novo transcriptome profiling of Naegleria fowleri trophozoites and cysts 2 by RNA-sequencing

This manuscript uses RNA seq to compare gene expression in the trophozoites and cyst stages of Naegleria fowleri the causative organism of a rare but usually fatal disease in humans (and other animals). The paper is well written and clear for the most part. The conclusions seems well supported by the described data. A few minor points are raised below.

Abstract Line 14.  It would be more correct to include the word “usually” before “fatal” as PAM has a mortality of around 96%. Also note that the word “fetal” here should be “fatal” as PAM affects all ages. (the authors correctly use the term “fatal” later in the introduction).

Line 29. This should be “On the other hand….” (singular)

Line 41. The abbreviation PAM should be used in line 41 where the term is used first in the main text of this paper instead of on line 49.

Line 55. Should be “amoeba” singular here not “amoebae” plural.

Line 63. This paragraph needs revision as only Schistosoma, Fasciola and Clonorchis are trematodes on this list. Also, a new paragraph is not needed on line 66 as the subject (transcriptomes) has not changed?

The figure 1 legend does not adequately explain this figure? It is not clear how the terms “UP DEG, Down DEG and SUM relates to each other.

Line 77. “into” not “tinto”

Line 183.  Although it is known that the profilin protein from amoebae such as Acanthamoeba binds monomeric actin and slows the elongation phase of actin polymerization of Acanthamoeba actin (Pollard Lab), I am sure that this has not yet been done with the proteins expressed in Naegleria. It would be a surprise if the behaviour of the two proteins from Naegleria do not also display this behaviour but the proteins differ significantly and so it cannot be taken that this is the case.  The sentence “Because the profilin gene regulates the nucleation rate of actin polymerization and the rate of filament elongation and reduces the concentration of F-actin, profilin plays an important role in the pathogenesis of N. fowleri.” Should be rewritten as “Because the profilin gene regulates the nucleation rate of actin polymerization in other organisms including Acanthamoeba (Tseng et al, 1984) and the rate of filament elongation and reduces the  concentration of F-actin, profilin is assumed to play an important role in the pathogenesis of N. fowleri also.”

Tseng, P.C., Runge, M.S., Cooper, J.A., Williams Jr, R.C. and Pollard, T.D., 1984. Physical, immunochemical, and functional properties of Acanthamoeba profilin. The Journal of cell biology98(1), pp.214-221.

Line 191. Are the authors sure that there is only one profilin gene in the Naegleria fowleri genome and how sure can they be that they have identified the correct gene. A blast search reveals two profilin candidates in Naegleria fowleri XP_044563896.1 and XP_044559054.1?
